# Charged Satellite Drop Avoidance in Electrohydrodynamic Dripping

**DOI:** 10.3390/mi10030172

**Published:** 2019-03-01

**Authors:** Lei Guo, Yongqing Duan, Weiwei Deng, Yin Guan, YongAn Huang, Zhouping Yin

**Affiliations:** 1State Key Laboratory of Digital Manufacturing Equipment and Technology, Huazhong University of Science and Technology, Wuhan 430074, China; hustgl@hust.edu.cn (L.G.); duanyongqing@hust.edu.cn (Y.D.); 2Department of Mechanics and Aerospace Engineering, Southern University of Science and Technology, Shenzhen 518055, China; dengww@sustc.edu.cn; 3School of Energy and Power Engineering, Huazhong University of Science and Technology, Wuhan 430074, China; yinguan@hust.edu.cn

**Keywords:** satellite drop, electrohydrodynamic jet printing, charge relaxation time

## Abstract

The quality of electrohydrodynamic jet (e-jet) printing is crucially influenced by the satellite drop formed when the primary drop detaches from the meniscus. If the satellite drop falls onto the substrate, the patterns on the substrate will be contaminated. The electric charge carried by the satellite drop leads to more complex satellite/meniscus interaction than that in traditional inkjet printing. Here, we numerically study the formation and flight behavior of the charged satellite drop. This paper discovered that the charge relaxation time (CRT) of the liquid determines the electric repulsion force between the satellite drop and meniscus. The satellite drop will merge with the meniscus at long CRT, and fail to merge and deteriorate the printing quality at short CRT. The simulations are adopted to discover the mechanism of generation and flight behavior of charged satellite drops. The results show that the critical CRT decreases with the dielectric constant of the liquid and the supplied flow rate. Namely, for small dielectric constant and fixed CRT, the satellite drop is less likely to merge with the meniscus, and for high flow rate, the satellite drop is prone to merge with the meniscus due to the delay of necking thread breakup. These results will help to choose appropriate parameters to avoid the satellite drop from falling onto the substrate.

## 1. Introduction

Electrohydrodynamic jet (e-jet) printing has received much attention recently due to its high printing resolution, ink compatibility, and process flexibility [1,2,3]. It can adopt nozzles much smaller than those used in traditional inkjet printing [4,5], and can work in versatile manners such as the cone-jet mode, dripping mode, or microdripping mode [6,7,8,9,10,11]. E-jet printing is suitable for a variety of inks and has found widespread applications in printed electronics [12,13,14,15], DNA microarrays [16], protein microarrays [17], photonic devices [18,19], 3D structures [20,21,22], solar cells [23,24], and others [25].

In cone-jet mode, the charge level of the printed drop may exceed the Rayleigh limit and modest evaporation of the drop will lead to Coulombic fission and blurry printed patterns when printing on dielectric substrates. In contrast, the dripping mode has lower charge level and Coulombic fission can be prevented. However, satellite drops may appear in the dripping mode. It is known that satellite drops are undesirable for inkjet printing as they may deteriorate the resolutions of printed patterns. Existing experimental and numerical studies on e-jet printing in dripping mode [26,27,28,29,30] mainly focused on the breakup of the primary drop instead of the subsequent satellite drop formation and trajectories. The satellite drop formation in e-jet printing is similar to that in traditional inkjet printing [31,32,33,34,35], except that the electric forces play an important role in the satellite drop formation and interaction between the satellite drop and meniscus. Huo et al. [36] visualized the satellite flight behaviors in electrohydrodynamic dripping mode through high-speed imaging and found that the satellite drop merged with the meniscus when the applied voltage was sufficiently low. The liquid used in the experiment was ethanol. However, the mechanism of this phenomenon was not reported.

The electric field and electric stress play important roles in the satellite formation and flight behavior. The charge relaxation time (CRT or ${\tilde{t}}_{e}=\varepsilon_{L}/K$), which is the ratio between the dielectric constant ($\varepsilon_{L}$) and the conductivity (K) of the liquid, has a big influence on the electric field distribution and surface charge along the drop surface during electrohydrodynamic dripping. CRT describes the time scale required for the surface charge responses to the change of electric field. Whether the satellite drop will merge or separate from the meniscus can be predicted by a critical CRT which also depends on the dielectric constant of the liquid and the supplied flow rate. Here, we investigated the effect of CRT on the satellite drop merging and separation. The paper is organized as follows. First, the numerical method and governing equations are introduced. Second, the influence of the CRT on the generation and flight of the satellite drop is described and discussed. Finally, the influence of dielectric constant and flow rate on the critical CRT as well as the underlying mechanisms are discussed.

## 2. Numerical Methods

Figure 1a shows the schematic of the experimental setup. The liquid was supplied by pushing the syringe by a syringe pump. The high voltage was applied between the nozzle and the substrate. The high voltage was generated by a signal generator and a high-voltage amplifier. A Taylor cone formed at the front of the nozzle under the applied voltage. Figure 1b shows two representative flight behaviors of satellite drops. In both cases, a long and thin liquid thread formed when the primary drop detached from the meniscus that suspended on the nozzle, and the breakup of this liquid thread led to the generation of the satellite drop. For the upper images, the satellite drop moved toward the meniscus and eventually merged with the meniscus. Thus, the satellite drop did not influence the printed patterns. For the lower images, the satellite drop moved downward and separated from the meniscus. The satellite drop may oscillate between the meniscus and the main drop due to the electric repulsion between the satellite drop and the meniscus/main drop. Finally, the satellite drop may deposit on the substrate and form unwanted marks. Therefore, to achieve clean and high-resolution e-jet printing in dripping mode, it is necessary to understand the mechanism of satellite drop merging and separation. As shown in Figure 1c, the surface of the drop suffers normal electric stress, tangential electric stress, and surface tension stress during the breakup process. The stresses influence the breakup process of the drop and the subsequent satellite flight behavior. Once the satellite drop forms, it suffers electric repulsion force from the meniscus and the main drop. Finally, the satellite drop may merge with the meniscus or separate from the meniscus.

Figure 1d shows the axisymmetric numerical domain and the boundary conditions of the model, which consists of a nozzle of radius $\tilde{R}$ kept at potential $\tilde{\phi }$ in a cylindrical domain of air. The no-slip boundary condition is applied at the inner wall of the nozzle. In all the expressions, (·) denotes the dimensionless form of a quantity and ($\tilde{\cdot }$) denotes the dimensional form of a quantity. The electrical potential is applied at the nozzle and the top boundary. The zero potential and the outflow boundary condition are applied at the bottom. The symmetric boundary conditions are set for the velocity and potential at the left boundary and the symmetric boundary condition is applied at the right boundary for the electric potential. The governing equations are non-dimensionalized for the simulation (see Methods). At the nozzle inlet, a parabolic velocity profile is applied,
(1)$$v=v_{0}\left(1-r^{2}\right)$$
Here, v is the axial component of the velocity, r is the radial coordinate. The magnitude of the velocity can be tuned by changing $v_{0}$. Both the positive and negative charge densities ($c^{\pm }$) are set as constant at the nozzle inlet. The dimensionless inner and outer radii of the nozzle are 1 and 1.25, respectively. The dimensionless nozzle-to-substrate distance is 20. We used *Gerris*, an open-source code based on volume of fluid method to simulate the electrohydrodynamic dripping process [30,37]. The physics are based on the leaky dielectric model [38].

The governing equations of the leaky dielectric model are established for electrohydrodynamic dripping as follows. The liquid is assumed to be incompressible and Newtonian, and the mass conservation equation can be expressed as
(2)∇·u=0
where u is the velocity of the flow. The liquid flow of e-jet printing is governed by the modified Navier-Stokes equation [30]:(3)$$\frac{\partial u}{\partial t}+u\cdot \nabla u=-\nabla p+Oh\cdot \nabla^{2}u+F_{e}-2H\delta_{s}n$$
where p is the pressure of the liquid, $F_{e}$ is the electric stress on the liquid, −2H represents the surface tension stress, $\delta_{s}$ is the Dirac function which indicates that the surface tension stress only exists at the surface of the liquid. The surface tension stress is transformed into a volume stress in the liquid by the continuum surface force (CSF) model. The electric stress can be expressed as(4)$$F_{e}=\rho_{e}E-\frac{1}{2}E^{2}\nabla \varepsilon_{r}$$
where $\rho_{e}$ is the volume charge density, E is the electric field. The first term on the right-hand side of the equation represents the electrostatic stress due to the free charge under electric field. The second term represents the electric polar stress caused by the polarity of the liquid under electric field.

The motion of the charge in the liquid is governed by
(5)$$c_{t}^{\pm }+\nabla \cdot \left(c^{\pm }u\right)=\nabla \cdot \left(D^{\pm }\nabla c^{\pm }\right)\mp \nabla \cdot \left(\Lambda^{\pm }c^{\pm }E\right)$$

The relation of the electric field, the electric potential, and the free charge is
(6)$$\nabla \cdot \left(\varepsilon_{r}E\right)=-\nabla \cdot \left(\varepsilon_{r}\nabla \phi \right)=\rho_{e}$$
(7)$$\rho_{e}=\frac{ez^{+}c^{+}+ez^{-}c^{-}}{\varepsilon_{0}\tilde{\phi }/{\tilde{R}}^{2}}$$
where $z^{\pm }$ are the valences of the positive and negative charge species, respectively. The conductivity of the liquid can be expressed as
(8)$$K=\omega^{+}z^{+}c^{+}e+\omega^{-}z^{-}c^{-}e$$
where $\omega^{\pm }$ are the mobilities of the positive and negative charge species, respectively, and e represents the elementary charge.

Figure 2 shows the comparison of the experiment and the simulation. The satellite drop position in the simulation is the same as in the corresponding picture of the experiment. It can be seen that the satellite drop in the simulation falls down as that in the experiment, but the falling speed of the satellite drop in the simulation is smaller than that in the experiment. Maybe this is caused by the inaccuracy in calculating the electric stress between the meniscus and the satellite drop in the simulation.

## 3. Results and Discussion

There are 10 parameters influencing the electrohydrodynamic dripping process [39]: $$\rho ,\;\gamma ,\;\varepsilon_{L},\;K,\mu ,\varepsilon_{G},\tilde{R},\;\tilde{L},\;\tilde{\phi },\;{\tilde{Q}}_{s}$$ where ρ is the density of the liquid, γ is the surface tension of the liquid, $\varepsilon_{L}$ is the dielectric constant of the liquid, K is the conductivity of the liquid, μ is the viscosity of the liquid, $\varepsilon_{G}$ is the dielectric constant of air, $\tilde{R}$ is the inner radius of the nozzle, $\tilde{L}$ is the nozzle-to-substrate distance, $\tilde{\phi }$ is the applied voltage, ${\tilde{Q}}_{s}$ is the supplied flow rate. After non-dimensionalization, there are six dimensionless numbers describing the electrohydrodynamic dripping process (see Appendix A): $$\varepsilon_{r},\;t_{e},\;Oh,\;L,\;\phi ,\;Q_{s}.$$This study only considers the flow with low viscosity (Oh=0.05). The dimensionless voltage ϕ is 6.

### 3.1. The Influence of CRT on the Flight of Satellite Drop

Figure 3 shows the motion of satellite drops for liquids with different CRTs. By altering the conductivity of the liquid, the CRT can be varied while keeping other dimensionless parameters constant. When the CRT is long enough, the satellite drop moves upward and merges with the meniscus (Figure 3a). When the CRT is short, the satellite first moves up and then goes down due to electric repulsion force (Figure 3b,c). The satellite drop goes farther away from the meniscus for liquid with smaller CRT within the same duration after breaking up from the bottom of the neck. Figure 3d shows the satellite drop position with time. The satellite drop goes upward and eventually merges with the meniscus for the liquid with CRT of $t_{e}$ = 2.5. The satellite drop first moves up and then goes down for the liquid with CRT of $t_{e}$ = 0.667 and $t_{e}$ = 0.333. In order to avoid unwanted deposition of satellites, liquids with longer CRT should be adopted. Therefore, in order to prevent the satellite drop from falling onto the substrate, the CRT of the liquid should be sufficiently long.

The electric repulsion force between the meniscus and the satellite drop is important for the satellite flight behavior. Since the satellite drop and the meniscus have the same type of charge, the meniscus will electrostatically repel the satellite drop and alter the satellite motion, and the satellite drop acceleration is:(9)$$a=\frac{E_{d}}{\rho }\cdot \frac{q_{d}}{V}$$
where V is the volume of the satellite drop, a is the deceleration rate of the satellite drop, $E_{d}$ is the average electric field on the satellite drop, $q_{d}$ is the charge on the satellite drop. Equation (9) suggests the deceleration rate mainly depends on the charge-to-volume ratio ($q_{d}/V$). Figure 4a shows the amount of charges carried by the satellite drops and the satellite drop volumes for liquids with different CRTs. It can be seen that the charges carried by the satellite drops decrease with CRT and increases a little when the charge relaxation time is long. The satellite drop volume increases with CRT. The charge-to-volume ratio decreases with CRT as shown in Figure 4b. Since the charge-to-volume ratio is small for the liquid with long CRT, the deceleration rate of the satellite drop is small. Thus, the satellite drop can merge with the meniscus for the liquid with long CRT.

The initial speed of the satellite drop upon its formation also has a significant influence on the satellite drop motion. If the initial upward speed of the satellite drop is high, it is prone to merge with the meniscus. The initial speed is 0.833, 0.308, and 0.132 for fluids with charge relaxation time of 2.5, 0.667, and 0.333, respectively. The initial speed is higher for the satellites with longer CRT. The initial position is closer to the meniscus for the liquid with longer CRT. The initial position and initial speed of the satellite drop are determined by the breakup process of the drop. The breakup process is influenced by the electric field and electric stress distribution along the drop surface. Figure 5 shows the evolution of the droplet shape for liquids with different CRTs during the electrohydrodynamic dripping.

The variation of the CRT affects the breakup of the drop due to the difference in the electric field and electric stress on the drop surface. Figure 6a,b show the isopotential lines for liquids with different CRTs at the moment of t=0. The meniscus and the primary drop are nearly equipotential, but the potential changes along the neck. Figure 6c shows the velocity distribution along the z-direction for liquids with different CRTs. The velocity of the liquid in the meniscus and the main drop are low, therefore the surface charge densities can reach the electrostatic equilibrium state. But the velocity in the neck is high and changes significantly along the neck especially at both ends of the neck. During the breakup, the neck shrinks and stretches more rapidly than the charge density redistribution, so the electrostatic equilibrium state cannot be reached. Figure 6d shows the potential distributions along the z-direction. Because the migration speed of the charges in the liquid with large CRT is low, the potential drops faster along the neck.

The decrease of the potential along the neck induces a large tangential electric field. The tangential electric field determines the electric stresses on the drop surface. The electric stress mainly exists at the surface of the liquid, and the stress balance at the surface of the liquid is [40]:(10)$$p+\tau_{n}=-2H$$
where $\tau_{n}$ is the normal electric stress, which is also called the electromechanical surface tension [26], −2H represents the surface tension stress, which would be $2/r_{0}$ for a sphere drop of radius $r_{0}$. The surface tension stress points into the liquid and the normal electric stress points out of the liquid. The surface tension stress drives the shrinking of the neck and breakup, while the normal electric stress counters the surface tension stress. The larger the normal electric stress, the slower the shrinking of the neck. The tangential electric stress is negligible compared with the normal electric stress. The normal electric stresses consist of the electrostatic stress ($\tau_{e}$) and polar stress ($\tau_{p}$) [41], and the dimensionless normal electric stress can be expressed as:(11)$$\tau_{n}=\tau_{e}+\tau_{p}$$
(12)$$\tau_{e}=q_{s}\cdot E_{n}$$
(13)$$\tau_{p}=\left(\varepsilon_{r}-1\right)\cdot E_{s}^{2}$$

The dimensionless electrostatic stress ($\tau_{e}$) is determined by the surface charge density ($q_{s}$) and normal electric field ($E_{n}$) on the drop surface and the dimensionless polar stress ($\tau_{p}$) is proportional to the square of electric shear field ($E_{s}$). Figure 7a shows the profiles and the coordinates of the drops of different CRTs. Figure 7b shows $E_{s}$ along the surface of the drops with different CRTs. $E_{s}$ is high at both ends of the neck. This is caused by the large velocity change at both ends of the neck as shown in Figure 6c. $E_{s}$ along the neck is stronger for the liquid with longer CRT. Figure 7c,d show the normal electric stresses, electrostatic stresses, and polar stresses along the drop surface for the liquid with CRTs of $t_{e}=2.5$ and $t_{e}=0.333$. The stresses are large along the surface of the neck but small along the surface of the main drop and the meniscus. For the liquid with large CRT, the polar stress is much larger than the electrostatic stress and the normal electric stress is nearly equal to the polar stress. For the liquid with small CRT, both the electrostatic stress and the electric polar stress are small along the neck. So the normal electric stress of the liquid with large CRT along the neck is larger than that of the liquid with small CRT. The electric normal stress reaches its maximum at the upper end of the neck for the liquid with high CRT. Therefore, the shrinking of the neck and breakup of the upper end of the neck are delayed by the large electric normal stress as shown in Figure 5. The length of the neck is larger for liquid with large CRT and the volume of the resulting satellite drop is larger.

### 3.2. The Effect of Dielectric Constant on the Critical CRT

The critical CRT for the transition of the two different satellite drop flight behaviors is also affected by the dielectric constant of the liquid. Figure 8a shows that the critical CRT decreases with the dielectric constant of the liquid. When the CRT is larger than the critical value, the satellite drop merges with the meniscus. When the CRT is smaller than the critical value, the satellite drop separates from both the meniscus and the primary drop. Figure 8b shows the position of the satellite drop with time for different dielectric constants. The satellite drop merges with the meniscus for low dielectric constant but separates from the meniscus for high dielectric constants. The initial position of the satellite drop for the liquid with high dielectric constant is closer to the meniscus and the initial speed of the satellite drop is also higher.

Figure 9a shows the correlation between satellite drop volume and satellite drop charge with the relative dielectric constant of the liquid, where the CRTs of the liquids are the same. It can be seen that both the volume and charge decrease with the dielectric constant of the liquid. Figure 9b shows the ratio of charge to volume for the liquids with different dielectric constants. The ratio of the charge to volume does not change with the dielectric constant of the liquid. Thus, the ratio of the charge to volume will not influence the deceleration rate of the satellite drop according to Equation (9). Figure 9c shows that the neck length decreases with the dielectric constant of the liquid at the moment the primary drop detaches from the neck. The satellite drop is farther away from the meniscus for the liquid with small dielectric constant, therefore the satellite drop is less likely to merge with the meniscus. Figure 9d shows the normal electric stress distribution along the drop surface. The normal electric stress along the neck is stronger for the liquid with small dielectric constant, thus the shrinking of the neck for liquid with small dielectric constant is slower, which leads to a longer neck.

### 3.3. The Effect of Supplied Flow Rate on the Critical CRT

Figure 10a,b shows the satellite drop formation process for different supplied flow rates. As the supplied flow rate increases, the upper end of the neck delays its breakup. However, when the CRT of the liquid is short enough, the satellite drop eventually separates from the primary drop and meniscus. Figure 10c shows the position of the satellite drop with time for different flow rates. When the flow rate is low, the satellite drop falls downward. When the supplied flow rate is high, the satellite drop moves upward and merges with the meniscus. For high flow rate, although the initial position of the satellite drop is low, the satellite drop is still close to the meniscus due to the large meniscus volume. So, the higher the supplied flow rate, the satellite drop is more prone to merge with the meniscus. Figure 11 shows that the critical CRT decreases with the supplied flow rate.

## 4. Conclusions

The dripping mode of E-jet printing is numerically investigated to reveal the influence of the CRT on the generation and flight of satellite drops. The results show that the satellite drop merges with the meniscus when the CRT of the liquid is long. The charge-to-volume ratio of the satellite drop is small for liquid with large CRT, so the electric repulsion between the satellite drop and the meniscus is weak. This promotes the merging of the satellite drop with the meniscus for liquid with large CRT. The electric potential decreases along the neck due to rapid stretching prior to breakup, which induces a large tangential electric field as well as large normal electric stress along the neck surface. The normal electric stress delays the breakup of the upper neck, which makes the initial speed of the satellite drop larger and the initial position closer to the meniscus. This also promotes the merging of the satellite drop with the meniscus. The numerical results also show that the critical CRT is affected by the dielectric constant of the liquid and the supplied flow rate. The critical CRT distinguishing satellite merging or separation decreases with dielectric constant. As the dielectric constant of the liquid increases, the neck becomes shorter at the moment the primary drop detaches from the neck, while the charge-to-volume ratio of the satellite drop remains unchanged. For high dielectric constant, the initial position of the satellite drop is closer to the meniscus, therefore the satellite drop is more prone to merge with the meniscus. As the supplied flow rate increases, the breakup of the upper end of the neck delays, which promotes the satellite drop to merge with the meniscus. The findings in this work will help to choose appropriate parameters to absorb the satellite drop and improve the resolution of e-jet printing in the dripping mode.

## Figures and Tables

**Figure 1 micromachines-10-00172-f001:**
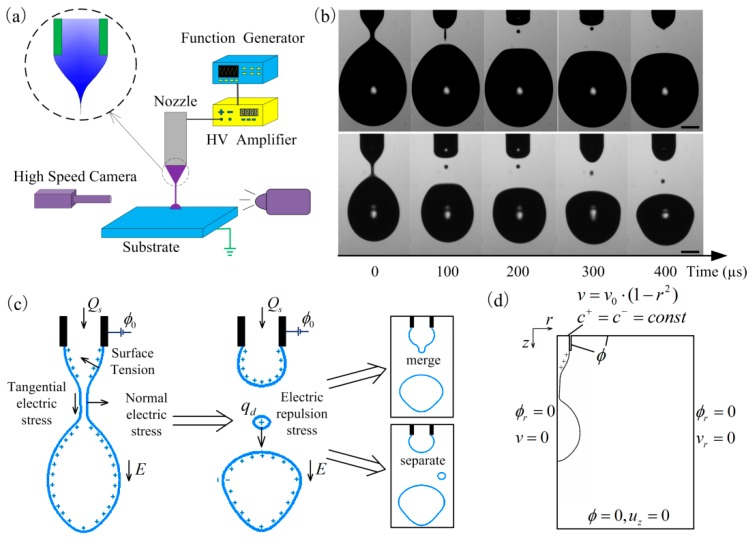
(**a**) The schematic of the experimental setup. (**b**) The two different satellite drop flight behaviors. For the upper images, the liquid was ethanol and the applied voltage was 1350 V. For the lower images, the liquid was a mixture of glycerin and water with the volume ratio of 1:2 and the applied voltage was 2300 V. The nozzle-to-substrate distance was 2.6 mm, which was 20 times the nozzle radius for both experiments. Scale bars: 200 μm. (**c**) The stress on the surface of the drop during breakup and the stress on the satellite drop during satellite drop motion. (**d**) The simulation domain and the boundary conditions for the simulation.

**Figure 2 micromachines-10-00172-f002:**
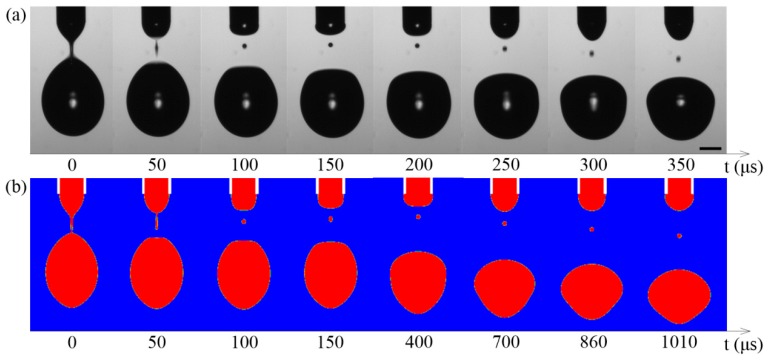
The comparison of (**a**) the experiment and (**b**) the simulation. The conductivity of the liquid is 2.3 μS/cm. The viscosity of the liquid is 3.5 mPa·s. The supplied flow rate is 223.5 μL/min. Scale bar = 100 μm.

**Figure 3 micromachines-10-00172-f003:**
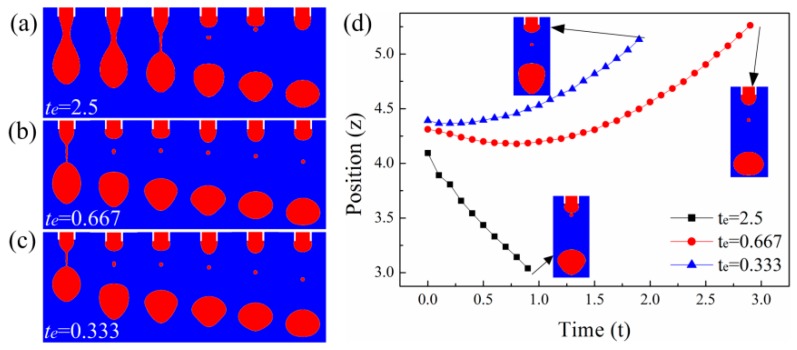
The satellite flight behaviors for liquids with CRTs of (**a**) $t_{e}=2.5$, (**b**) $t_{e}=0.667$, (**c**) $t_{e}=0.333$. The dimensionless time between adjacent images is 1. (**d**) The position of the satellite drops with time for liquids with different CRTs. The other parameters are Oh=0.05, ϕ=6, $Q_{s}=0.314$, $\varepsilon_{r}=50$.

**Figure 4 micromachines-10-00172-f004:**
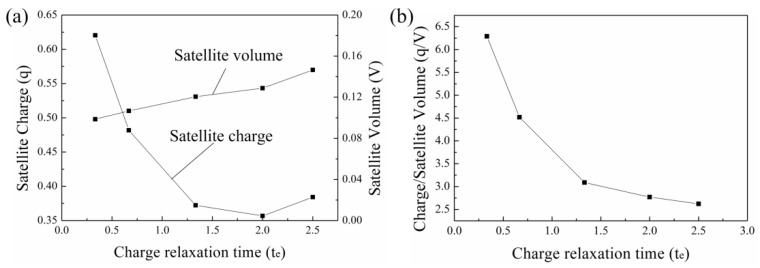
The satellite drops and their flight behavior with different CRTs. (**a**) The correlation between the satellite drop charge and CRT and the correlation between satellite drop volume and CRT. (**b**) The correlation between the charge-to-volume ratio of the satellite drop and CRT. The other parameters are Oh=0.05, ϕ=6, $Q_{s}=0.314$, $\varepsilon_{r}=50$.

**Figure 5 micromachines-10-00172-f005:**
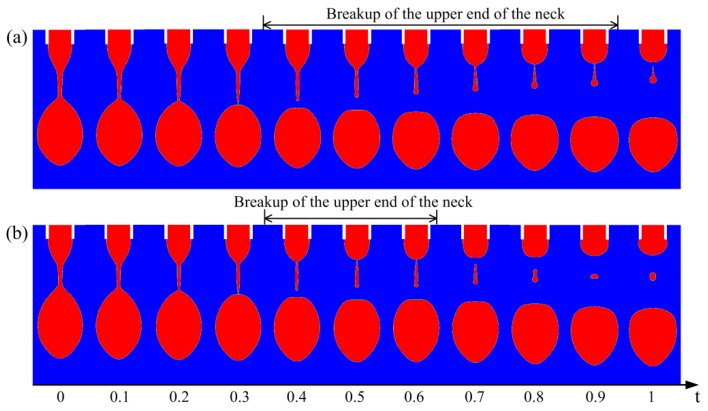
The evolution in time of the drop shape for liquids with CRTs of (**a**) $t_{e}=2.5$ and (**b**) $t_{e}=0.333$. The other parameters of the liquids are $\varepsilon_{r}=50$, Oh=0.05, ϕ=6, $Q_{s}=0.314$.

**Figure 6 micromachines-10-00172-f006:**
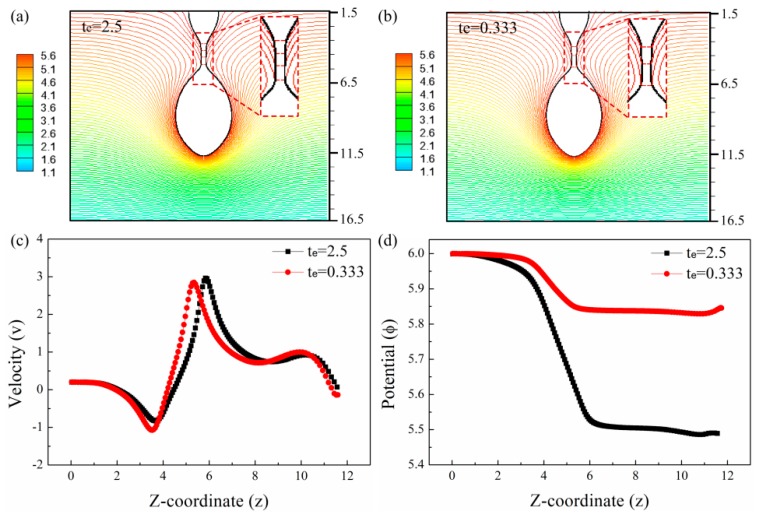
The isopotential lines at the instant when t=0 for liquid with CRT of (**a**) $t_{e}=2.5$ and (**b**) $t_{e}=0.333$. (**c**) The velocity distribution for liquids with different CRTs at the instant when t=0. (**d**) The potential distribution for liquids with different CRTs at the instant when t=0. The other parameters of the liquids are $\varepsilon_{r}=50$, Oh=0.05, ϕ=6, $Q_{s}=0.314$.

**Figure 7 micromachines-10-00172-f007:**
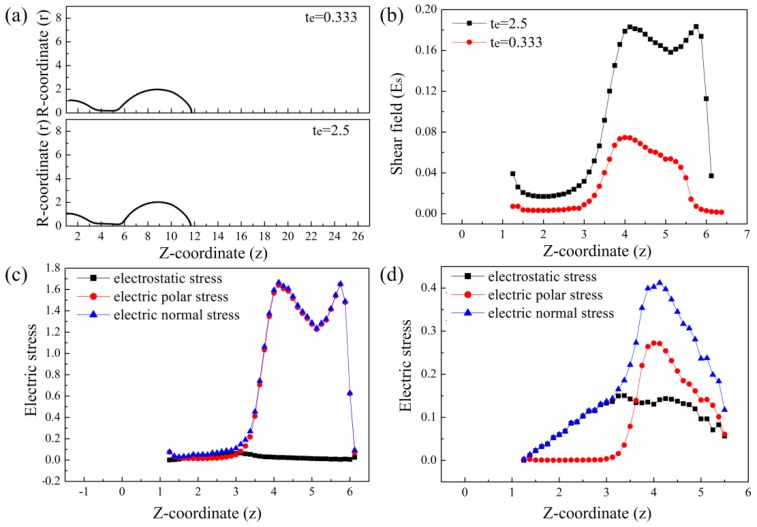
The comparisons of the electric stresses for liquids with different CRTs at the instant when t=0. (**a**) The profiles of the drops of liquids with different CRTs. (**b**) The electric shear field along the surface of the drop for liquids with different CRTs. (**c**) The electric normal stress for liquid with CRT of $t_{e}=2.5$. (**d**) The electric normal stress for liquid with CRT of $t_{e}=0.333$. The other parameters of the liquids are $\varepsilon_{r}=50$, Oh=0.05, ϕ=6, $Q_{s}=0.314$.

**Figure 8 micromachines-10-00172-f008:**
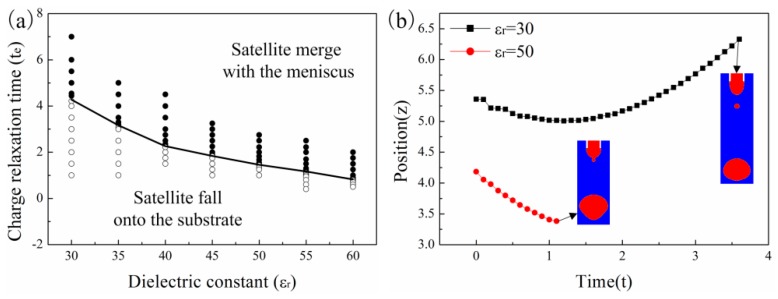
The satellite drop flight behavior for liquids with different dielectric constants. (**a**) The correlation between the critical CRT for the different satellite flight behaviors and dielectric constant. (**b**) The position of the satellite drop with time for liquids with CRT of $t_{e}=2$. The other parameters are Oh=0.05, ϕ=6, $Q_{s}=0.314$.

**Figure 9 micromachines-10-00172-f009:**
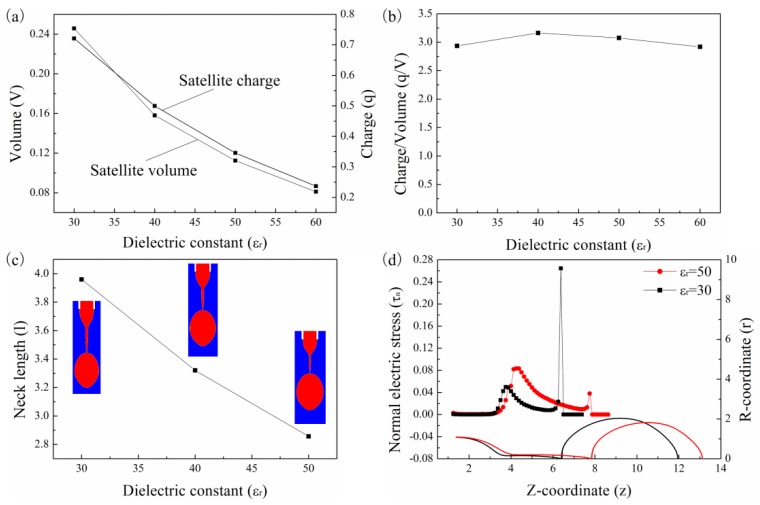
(**a**) The correlation of the satellite drop volume and satellite drop charge with dielectric constant. (**b**) The correlation between the charge-to-volume ratio of the satellite drop and dielectric constant. (**c**) The neck length for liquids with different dielectric constants. (**d**) The normal electric stress for liquids with different dielectric constants. The other parameters are Oh=0.05, ϕ=6, $Q_{s}=0.314$, $t_{e}=2$.

**Figure 10 micromachines-10-00172-f010:**
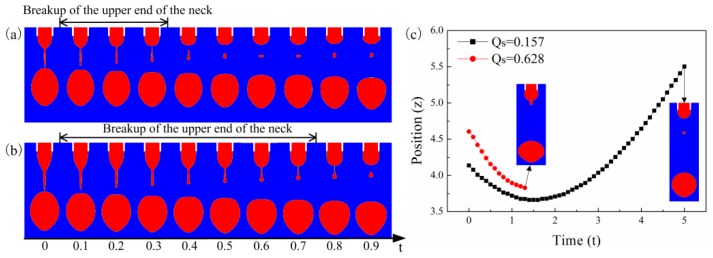
The satellite drop formation for supplied flow rate of (**a**) $Q_{s}=0.157$ and (**b**) $Q_{s}=0.628$. (**c**) The satellite drop position with time for different supplied flow rates. The other parameters are $\varepsilon_{r}=50$, Oh=0.05, ϕ=6, $t_{e}=1.429$.

**Figure 11 micromachines-10-00172-f011:**
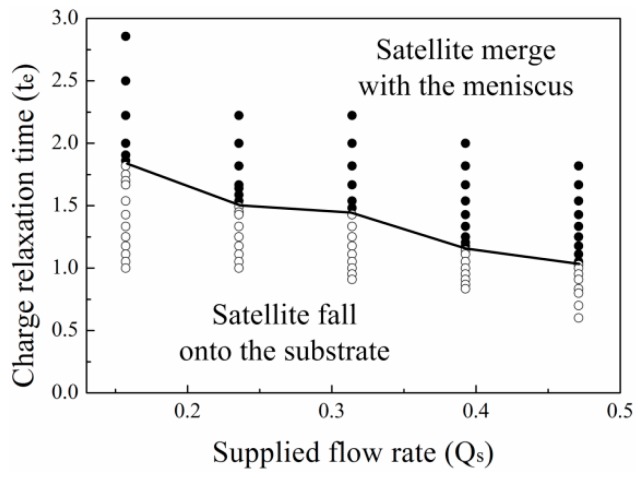
The critical CRT for the different satellite flight behaviors with different supplied flow rates. The other parameters are $\varepsilon_{r}=50$, Oh=0.05, ϕ=6.

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
