# Peer review of "Charged Satellite Drop Avoidance in Electrohydrodynamic Dripping"

_micromachines, 2019, doi:10.3390/mi10030172_

Round 1

Reviewer 1 Report

Dear authors,

it would improve the paper if you report state-of-the-art in terms of satellite avoidance for electrohydrodynamic printing;

the CCD is only useful to determine quadratic models, so I would suggest to revise this part and restrict yourself to a model with (much) less coefficients; - what is the increase in the error if you do so?

I further think that 30 experiments are much too little to determine 16 model coefficients, as a rule of thumb the number of experiments should be 10x number of coefficients; since you do 'numerical' experiments, this should be easy to achieve;

two experiments also seem pretty little to validate the described model! please add some more evaluations here and present also the standard deviation (e.g. whisker plots)

Best wishes

Author Response

We would like to express our sincere thanks to you for your review. The point-to-point response to your comments is in the following word profile.

Reviewer 2 Report

While the manuscript is generally well presented and argued, the fundamental area for improvement is that it is not apparent from the manuscript that the modelling work has been validated against any experimental observation (either performed by the authors or based on pre-existing literature). While the conclusions made by the authors seem plausible, without any validation against experimental results it is difficult to determine whether these conclusions are relevant to real world conditions or an artefact of their modelling.

Author Response

Thank you for your review.The point-to-point response of your comments is in the following word profile.

Round 2

Reviewer 2 Report

Accept